# Sociodemographic characteristics, community engagement and stigma among Men who have Sex with Men (MSM) who attend MSM-led versus public sexual health clinics: A cross-sectional survey in China

**Christina Huon** [1], **Gifty Marley** [2], **Rayner Kay Jin Tan**[2], **Dan Wu**[3,4], **Qilei Sheng**[2], **Ye Liu**[2], **Margaret Elizabeth Byrne**[5], **Qiwen Tang**[2], **Rong Mu**[2], **Cheng Wang**[6], **Ligang Yang**[6], **Tong Wang**[2], **Weiming Tang**[7], **Joseph D. Tucker** [3,7] *

1 Newcastle University, Newcastle, England, United Kingdom, 2 University of North Carolina Project–China, Guangzhou, China, 3 Clinical Research Department, London School of Hygiene and Tropical Medicine, London, United Kingdom, 4 School of Public Health, Nanjing Medical University, Nanjing, China, 5 University of North Carolina at Chapel-Hill, Chapel Hill, NC, United States of America, 6 Dermatology Hospital, Southern Medical University, Guangzhou, China, 7 Department of Medicine, University of North Carolina at Chapel-Hill, Chapel Hill, NC, United States of America

* jdtucker@med.unc.edu

## Abstract

Community-based sexual health services are recommended to increase sexually transmitted disease (STD) testing among men who have sex with men (MSM). Pilot study data from multiple sites found that MSM in Guangzhou who use public STD clinics were found to have different sociodemographic characteristics, lower community engagement, and increased social cohesion, compared to MSM who use MSM-led clinics.

## Introduction

Chlamydia and gonorrhoea are common among men who have sex with men (MSM) in China, with testing rates as low as 28% for Chlamydia and 30% for Gonorrhoea [1]. HIV testing rates among Chinese MSM are much higher at approximately 79%, and this difference can be attributed to national MSM HIV testing programs [1]. Currently, there are no specific guidelines for Chlamydia and Gonorrhoea testing among MSM in China.

Most Chinese people access healthcare through public hospitals. The financial cost of seeking healthcare is subsidized through health insurance systems, such as basic medical insurance provided by the Chinese government, and commercial insurance through private companies [2]. HIV testing is subsidised by national epidemic prevention and control programmes but does not include free testing for other STDs [3]. Furthermore, most health insurance packages do not cover routine testing and treatment for common STDs, leading to out-of-pocket payments of approximately $22 USD [4]. Despite this cost, public STD clinics are essential in providing greater privacy for closeted MSM, whose attendance at an MSM-led clinic could potentially expose an individual's sexual behaviours and sexual identity.

**Data Availability Statement:** All relevant data are within the paper and its Supporting Information files.

**Funding:** National Institutes of Health (Grant number NIH NIAID R01AI158826) The funders had no role in study design, data collection and analysis, decision to publish, or preparation of the manuscript.

**Competing interests:** The authors have declared no competing interests exist.

The WHO recommends implementation of community-based sexual health services, such as MSM-led clinics, to increase STI testing among key populations, including MSM [5]. MSM-led clinics are staffed by MSM volunteers, offering MSM-friendly services specifically for MSM users.

Chinese non-governmental organizations (NGOs), including community-based organizations (CBOs) develop these services to supplement the healthcare needs of their communities. Individuals who seek care at MSM-led clinics report feeling more supported by staff and receiving care which prioritized confidentiality more than public STD clinic services [6–8].

However, there are also disadvantages to MSM seeking sexual health services at MSM-led clinics. Although same-sex behaviours were decriminalized in China in 1997 and declassified as part of the Chinese Classification of Mental Disorders in 2001 [9, 10], there remains a lack of legal protection for MSM against discrimination, and same-sex marriage is illegal. Stigma against MSM persists among the population, including healthcare staff [7]. Chinese MSM face many socio-cultural barriers to STD testing, such as stigma, rejection, and fear of bringing shame on an individual's family due to their sexuality [11].

Previous studies have investigated the effect of sociodemographic characteristics, testing status, and testing barriers on sexual health outcomes in Chinese MSM [4, 12, 13], but few studies have examined the characteristics of MSM who seek care in different settings. Some studies have investigated the psychological characteristics of MSM in China and one found that MSM with and without high-risk homosexual behaviors in China have different social and psychological characteristics [14–16]. However, none of these studies have assessed the relationship between these psychological characteristics and healthcare seeking behaviors and testing sites among MSM in China to our knowledge. Additionally, most studies of healthcare seeking behaviour have focused on high income countries [17]. The limited literature on health seeking behaviours has also not addressed community engagement, homonegativity and related social forces that likely influence clinic choice [18]. The purpose of this study was to compare the sociodemographic and psychosocial characteristics of MSM who sought clinical services at MSM-led clinics compared to public STD clinics using data from a cross-sectional study. This analysis aimed to inform a contextualized trial design and provides an important initial understanding of two types of mainstream clinic providing STD testing services for MSM in China.

## Methods

This data was collected as part of pilot studies in Guangzhou, China from 22 May 2022 to June 2023 [19]. A convenience sample of MSM were enrolled from 14 clinics (7 MSM-led and 7 public STD clinics) in seven cities in Guangdong province. Both types of clinic operated in the same facility in three cities (Guangzhou, Foshan, and Jiangmen) and in facilities independent of each other in four cities (Zhuhai, Huizhou, Zhanjiang, and Dongguan). The two types of facilities were similar in their use of online appointment booking systems with options for walk-in clients at the public STD clinic in case of emergencies but differed in working hours as MSM-led clinics provided services tailored by the times convenient to their clients (including night time testing, weekend only services, and targeted onsite testing at MSM social gathering events). The MSM-led clinics were staffed by nurses and MSM volunteers, and strongly focused on point of care STD screening, counselling and referral. The public STD clinics, staffed by physicians and nurses, provided the clinical standard of care, including STD screening, counselling, diagnostic testing, treatment and vaccination. S1 Fig provides more details on the similarities and differences between the two clinic types. These differences provided a unique opportunity to compare health seeking characteristics more directly among MSM choosing services at MSM-led or public clinics.

Inclusion criteria for this study included: age greater than 18 years old; ever had anal sex with another man; resided in Guangdong Province for the past 3 months; spoke Mandarin Chinese or Cantonese; mentally capable of providing informed consent; and owned a mobile phone. Men who had been tested for gonorrhoea or chlamydia in the last 12 months, or were on treatment for chlamydia, gonorrhoea or other STDs were excluded.

### Ethical approval

The study was approved by the ethical review boards of the University of North Carolina at Chapel Hill and the Dermatology Hospital of Southern Medical University. All eligible participants provided informed consent online by selecting "I agree" or "I disagree" after reading the full informed consent statement. Only participants who consented by selecting "I agree" were redirected to complete the self-administered survey.

### Data collection

Data on sociodemographic information and psychosocial characteristics were collected via a self-administered online survey, completed in the clinic after individuals had consented to participate in the study [see S1 File].

Each psychosocial characteristic was measured using average sum of item statements ranked by Likert scale, with statements adapted from previous studies [20]. Psychosocial factors measured included: community engagement, community connectedness, social cohesion, internalized homonegativity and perceived stigma. Community engagement consisted of five items, which measured the level at which an individual participates in the MSM community. Community connectedness referred to how an individual felt in relation to the MSM community, including a sense of belonging and shared goals [21]. Social cohesion was defined as the strength of relationships, solidarity and trust between members of the MSM community. Internalized homonegativity was measured using nine items, based on negative feelings and homonegative attitudes an MSM may feel towards themselves or other sexual minorities [22].

### Statistical analysis

A descriptive analysis was conducted to summarize the sociodemographic characteristics, sexual behaviors, and STD testing experience of participants. Sociodemographic data were analyzed for inter-group differences using Pearson's chi-squared test or Fisher's exact test. The association between psychosocial factors and clinic presentation was examined using univariate and multivariable logistic regression. Socio-demographics, psychosocial characteristics, and clinic type (public STD vs. MSM-led). Variables found to be marginally associated (a priori determined as $p<0.20$) with clinic type in crude bivariate analysis were included in a multivariable logistic regression model. No variable had >15% missing data and variables with missing data were exempted from the regression models using a listwise deletion approach. Age, level of education, and marital status were controlled in the regression model as confounders of the clinic type. The results were reported as adjusted odds ratio(aOR) with 95% confidence intervals (C.I) and p-values [23].

## Results

Two hundred and thirty-eight men completed the survey questionnaire. Table 1 summarizes the socio-demographic and psychosocial characteristics of the participants. The overall mean age was 29 (±8.6) years old. Most of the participants had never been married (196/238, 82.4%), and the majority have ever had a STD test (217/283, 94.3%).

**Table 1. Comparison of sociodemographic, sexual behaviour and STD testing characteristics of MSM at 14 sites, participating in a pilot study in China, 2022.** N = 238(%).

| Sociodemographics | Total, N = 238 | Public STD, n = 63 | MSM-led, n = 175 | *P*-value |
|---|---|---|---|---|
| **Age (years)/ mean±S.D** | 29±8.60 | 27±6.76 | 31±8.97 | 0.006[1] |
| **18–24** | 75 (31.5) | 26 (41.3) | 49 (28.0) | |
| **25–29** | 66 (27.7) | 22 (34.9) | 44 (25.1) | |
| **30–34** | 48 (20.2) | 11 (17.5) | 37 (21.1) | |
| **35+** | 48 (20.2) | 4 (6.3) | 44 (25.1) | |
| **Marital Status** | | | | 0.011[1] |
| **Never married** | 196 (82.4) | 59 (93.7) | 137 (78.3) | |
| **Ever Married** | 42 (17.6) | 4 (6.3) | 38 (21.7) | |
| **Education level** | | | | 0.001[1] |
| **Bachelor's degree and below** | 65 (27.3) | 7 (11.1) | 58 (33.1) | |
| **Above Bachelor's** | 173 (72.7) | 56 (88.9) | 117 (66.9) | |
| Annual Income (USD[#]) | | | | 0.154[1] |
| **<415.85** | 57 (23.9) | 20 (31.7) | 37 (21.1) | |
| **415.85–693.08** | 66 (27.7) | 13 (20.6) | 53 (30.3) | |
| **>693.08** | 115 (48.3) | 30 (47.6) | 85 (48.6) | |
| **Number of sexual partners in last 12 months** | | | | 0.104[1] |
| One | 117 (49.2) | 37 (58.7) | 80 (45.7) | |
| 2+ | 121 (50.8) | 26 (41.3) | 95 (54.3) | |
| **Ever engaged in condomless sex in past 6months** | | | | <0.001 |
| **Yes** | 152 (63.9) | 33 (52.4) | 119 (68.0) | |
| **No** | 86 (36.1) | 30 (47.6) | 56 (32.0) | |
| Ever tested for STD [+] | | | | 0.008[2] |
| Yes | 217 (94.3) | 47 (85.5) | 170 (97.1) | |
| No | 12 (5.2) | 7 (12.7) | 5 (2.9) | |

Note

[1]Chi-Squared test

[2]Fisher's Exact test

[#]USD = United States Dollars

[+]STD: HIV, Syphills, Hepatitis B, Hepatitis C, Gonorrhea, Chlamydia; Ever married included participants who were currently married, divorced, separated, and widowed.

The overall measure for each psychosocial factor was categorized using a ternary of 0.33 and 0.66 around the mid-point of each scale, where average scores <0.33 = low, 0.33–0.66 = moderate, and >0.66 = high. Social cohesion was lower in MSM seeking care at the MSM-led clinic (aOR = 0.42, 95% C.I: 0.19–0.92; p = 0.031). Community engagement was also higher in MSM presenting at the MSM-led clinic (aOR = 3.84, 95%C.I:1.59–9.25; p = 0.003). Further details are available in Table 2.

## Discussion

This study sought to compare MSM who sought sexual health services at an MSM-led clinic to MSM who sought care at a public STD clinic. Our findings show that sociodemographic characteristics (age, level of education, income, marital status) and history of STD testing between the two groups of MSM differ, but psychosocial factors did not differ largely. This extends the limited literature on factors predicting health seeking behaviours among MSM in China,

**Table 2. Logistic regression outcomes showing the association between psychosocial characteristics of MSM at 14 sites participating in a pilot study in China, and the choice of clinic type for STD services, 2022.** *N = 238(%).*

| Psychosocial factors | Public clinic, n = 63 (%) | MSM clinic, n = 175(%) | OR(95%CI) | AOR(95%CI) |
|---|---|---|---|---|
| Community Engagement | | | | |
| Low | 27(42.9) | 52(29.7) | 0.93(0.49–1.78) | 0.92 (0.47–1.79) |
| Middle | 28(44.4) | 58(33.1) | Ref | Ref |
| High | 8(12.7) | 65(37.1) | 3.92(1.66–9.29)** | 3.84 (1.59–9.25)** |
| Community Connectedness | | | | |
| Low | 15(23.8) | 41(23.4) | 0.97(0.46–2.03) | |
| Middle | 27(42.9) | 76(43.3) | Ref | |
| High | 21(33.3) | 58(33.1) | 0.98(0.50–1.91) | |
| Social Cohesion | | | | |
| Low | 27(42.9) | 57(32.6) | 0.46(0.23–0.92)** | 0.61 (0.30–1.24)$^\beta$ |
| Middle | 18(28.6) | 82(46.9) | Ref | ref |
| High | 18(28.6) | 36(20.6) | 0.44(0.20–0.94)** | 0.42 (0.19–0.92)** |
| Internalized Homonegativity | | | | |
| Low | 22(34.9) | 46(26.3) | 0.59(0.29–1.20) | |
| Middle | 20(31.7) | 74(42.3) | Ref | |
| High | 20(34.9) | 55(31.4) | 0.78(0.39–1.58) | |

**Note**: OR = odds ratio; aOR = adjusted odds ratio; CI = confidence interval

**P<0.05

*P<0.1

$^\beta$p<0.2

leverages on a unique opportunity in which two services are provided within the same space, and explores the social factors known to drive STD testing behaviours.

We observed significant differences in sociodemographic characteristics (age, marital status, and education level) and history of STD testing between MSM attending the MSM-led clinic and those at the public STD clinic. This is inconsistent with findings from one study in Africa, but concurs with a previous multi-center study in China [24, 25]. More MSM at the MSM-led clinic had never been married at time of testing compared to MSM at the public STI clinic, but this can mostly be explained by the local cultural phenomenon of wives escorting their spouses to seek healthcare services, and attending the MSM-led STD clinic could lead to sexual identity disclosure.

Our data suggest that the participants in MSM-led clinics are thrice more likely to have a high community engagement compared to MSM at the public STD clinic (aOR = 3.84, 95%C. I:1.59–9.25; p = 0.003). MSM recruited from MSM-led clinics may be more interested in research engagement, volunteering or other activities because of their relatively greater community engagement experiences. This study extends the literature from other countries which suggest that MSM who prefer public STD clinics are more likely to have lower levels of community engagement and social cohesion [6, 24, 26]. However, we also observed that MSM presenting to MSM-led clinics are less likely to have a high level of social cohesion (aOR = 0.42, 95%C.I: 0.19–0.92; p = 0.031) compared to MSM in public STD clinics. This contradicts findings from a study in Myanmar [26]. The low engagement among MSM at the public STD clinic show that there is need for more widespread community engagement interventions outside of CBOs.

We observed a trend toward less internalized homonegativity among MSM seeking care in the MSM-led clinic compared to the public STD clinic. Higher levels of internalized

homonegativity have been associated with lack of sexual behavior disclosure to healthcare staff, which in turn can lead to omission of appropriate tests such as rectal swabs [6, 27]. This could be partly explained by the higher levels of community engagement we found among MSM-led clinic participants, which indicate a sense of belonging to the MSM community and acceptance by the MSM community [15, 28, 29]. If an individual was ashamed of their sexuality, or did not feel part of the MSM community, it is reasonable that they would prefer to attend a public STD clinic, rather than an MSM-specific clinic.

This study has several limitations. First, this was a small sample, limiting the extent to which inferences can be made on this data. Secondly, these data are from a single point in time and do not reflect temporal trends. Third, the MSM-led clinics did not offer STD treatment. Therefore, MSM who were symptomatic may have chosen to seek care at the public STD clinic instead. Furthermore, the data on sexual behaviors and psychosocial characteristics are self-reported and may be subject to recall and social desirability bias. Our investigation addressed sensitive questions that may have led to the misreporting of personal sexual risk and STD testing behavior data owing to social desirability bias. Therefore, our findings and the conclusions drawn should be interpreted with caution. Finally, all the study participants were recruited from an urban areas with a well-established CBO. Whilst this is not likely to reflect MSM in rural areas, there are more MSM-led clinics in similar urban areas.

## Conclusion

This study extends previous research which found patterns of subgrouping among MSM with different preferences for self-testing and choice of care site. Although further research is needed to better define these two subgroups, our findings suggest that MSM who attend public STD clinics tended to have lower levels of community engagement with the local LGBTQ + community and greater internalized homonegativity compared to MSM who attend MSM-led clinics. Consequently, hypothetical targeted interventions implemented solely through MSM-led community organizations may not be adequate to reach them. Public STD clinics in China could be considered as a site to deliver anti-stigma interventions to MSM. These data suggest that co-location of MSM-led services on top of existing public STD clinics help to reach MSM who would not be usually reached by public clinics.

## Supporting information

**S1 Fig. Infographic showing the differences and similarities between the two types of clinics attended by MSM in China.**
(TIF)

**S1 File. Study questionnaire used for data collection.**
(PDF)

**S1 Dataset.**
(XLSX)

## Acknowledgments

Special thanks to the staff of the Dermatology Hospital of the Southern Medical University, staff of all our public STD clinics and MSM-led community-based organizations partners for their contribution in designing and implementing this study. We are also grateful to all MSM participants for their voluntary participation.

## Author Contributions

**Conceptualization:** Gifty Marley, Rayner Kay Jin Tan, Dan Wu, Margaret Elizabeth Byrne, Cheng Wang, Ligang Yang, Weiming Tang, Joseph D. Tucker.

**Data curation:** Gifty Marley, Qilei Sheng, Ye Liu, Qiwen Tang, Rong Mu, Tong Wang.

**Formal analysis:** Christina Huon, Gifty Marley.

**Investigation:** Gifty Marley, Rayner Kay Jin Tan, Cheng Wang, Ligang Yang, Weiming Tang, Joseph D. Tucker.

**Methodology:** Rayner Kay Jin Tan, Margaret Elizabeth Byrne, Joseph D. Tucker.

**Project administration:** Gifty Marley, Rayner Kay Jin Tan, Qilei Sheng, Ye Liu, Qiwen Tang, Rong Mu, Cheng Wang, Ligang Yang, Tong Wang.

**Supervision:** Joseph D. Tucker.

**Writing – original draft:** Christina Huon, Gifty Marley.

**Writing – review & editing:** Rayner Kay Jin Tan, Dan Wu, Qilei Sheng, Ye Liu, Joseph D. Tucker.

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
