## [Decision Letter · Decision Letter 0]

24 Jul 2024

PONE-D-24-26827Sociodemographic characteristics, community engagement and stigma among Men who have Sex with Men (MSM) who attend MSM-led versus public sexual health clinics: A cross-sectional survey in ChinaPLOS ONE

Dear Dr. Marley,

Thank you for submitting your manuscript to PLOS ONE. After careful consideration, we feel that it has merit but does not fully meet PLOS ONE’s publication criteria as it currently stands. Therefore, we invite you to submit a revised version of the manuscript that addresses the points raised during the review process.

We look forward to receiving your revised manuscript.

Kind regards,

Geng-Feng Fu, Ph.D.

Academic Editor

PLOS ONE

Journal Requirements:

2. Please include a complete copy of PLOS’ questionnaire on inclusivity in global research in your revised manuscript. Our policy for research in this area aims to improve transparency in the reporting of research performed outside of researchers’ own country or community. The policy applies to researchers who have travelled to a different country to conduct research, research with Indigenous populations or their lands, and research on cultural artefacts. The questionnaire can also be requested at the journal’s discretion for any other submissions, even if these conditions are not met.  

Please find more information on the policy and a link to download a blank copy of the questionnaire here: https://journals.plos.org/plosone/s/best-practices-in-research-reporting. 

Please upload a completed version of your questionnaire as Supporting Information when you resubmit your manuscript.

"National Institutes of Health (Grant number NIH NIAID R01AI158826)"

4. In the online submission form you indicate that your data is not available for proprietary reasons and have provided a contact point for accessing this data. Please note that your current contact point is a co-author on this manuscript. According to our Data Policy, the contact point must not be an author on the manuscript and must be an institutional contact, ideally not an individual. Please revise your data statement to a non-author institutional point of contact, such as a data access or ethics committee, and send this to us via return email. Please also include contact information for the third party organization, and please include the full citation of where the data can be found.

**Additional Editor Comments:**

Reviewer 1:

his research article provides significant insights for the public health field by comparing the sociodemographic and psychosocial characteristics of Chinese men who have sex with men (MSM) seeking sexual health services at different types of clinics. The study targets the MSM population, a crucial group for the prevention and control of sexually transmitted diseases (STDs), which holds global significance in public health. Overall, this paper holds substantial research value in promoting the optimization of sexual health services for MSM in China. Some minor problems should be revised for the current version manuscript.

Introduction:

It is recommended to provide more background information on the existing literature and knowledge gaps regarding health-seeking behaviors and psychosocial factors among MSM in China.

Methods:

It is suggested to provide more details on the statistical analysis, including the variable selection process for the multivariate models and the handling of potential confounding factors.

Statistical analysis:

Further details should be provided on the statistical analysis, including the variable selection process for the multivariate models and the handling of potential confounding factors.

Results:

The current Table 1 on the comparison of sociodemographic and psychosocial characteristics of MSM is somewhat disorganized; partitioning it into two tables would be more appropriate.

Discussion:

It is recommended to discuss the potential limitations of the self-reported data and their impact on the validity of the study results.

Reviewer 2:

This study was found that MSM in Guangzhou who use public STD clinics were found to have different sociodemographic characteristics, lower community engagement, and increased social cohesion, compared to MSM who use MSM-led clinics. Some comments and suggestions are as follows:

1. In the “Intrdution”, P.3”MSM-led clinics. Although same-sex behaviours were decriminalized in China in 1997, thereremains a lack of legal protection for MSM against discrimination, and same-sex marriage is illegal”, please provide references.

2. There was a big difference in sample size between the two groups, Public STD is only 63 participants and MSM-led is 175 participants. Was the sample size enough for this study?

3. In Table 1, it is suggested that the analysis of the two groups of sociodemographic characteristics, and psychological factors related to attend clinics, should not be placed in the same table.

4. Why did both types of clinic operate in the same facility in three cities (Guangzhou, Foshan, and Jiangmen)?

5. Was there any information collected on why the subjects chose to visit these institutions?

6. In Table 1, it is recommended to supplement subjects' condom use during sexual behaviors.

7. It is suggested that more detailed information on the two different types of institutions be added to the method.

Reviewers' comments:

Reviewer's Responses to Questions

**Comments to the Author**

1. Is the manuscript technically sound, and do the data support the conclusions?

Reviewer #1: Yes

Reviewer #2: Yes

2. Has the statistical analysis been performed appropriately and rigorously? 

Reviewer #1: Yes

Reviewer #2: Yes

3. Have the authors made all data underlying the findings in their manuscript fully available?

Reviewer #1: No

Reviewer #2: No

4. Is the manuscript presented in an intelligible fashion and written in standard English?

Reviewer #1: Yes

Reviewer #2: Yes

5. Review Comments to the Author

Reviewer #1: This research article provides significant insights for the public health field by comparing the sociodemographic and psychosocial characteristics of Chinese men who have sex with men (MSM) seeking sexual health services at different types of clinics. The study targets the MSM population, a crucial group for the prevention and control of sexually transmitted diseases (STDs), which holds global significance in public health. Overall, this paper holds substantial research value in promoting the optimization of sexual health services for MSM in China. Some minor problems should be revised for the current version manuscript.

Introduction:

It is recommended to provide more background information on the existing literature and knowledge gaps regarding health-seeking behaviors and psychosocial factors among MSM in China.

Methods:

It is suggested to provide more details on the statistical analysis, including the variable selection process for the multivariate models and the handling of potential confounding factors.

Statistical analysis:

Further details should be provided on the statistical analysis, including the variable selection process for the multivariate models and the handling of potential confounding factors.

Results:

The current Table 1 on the comparison of sociodemographic and psychosocial characteristics of MSM is somewhat disorganized; partitioning it into two tables would be more appropriate.

Discussion:

It is recommended to discuss the potential limitations of the self-reported data and their impact on the validity of the study results.

Reviewer #2: This study was found that MSM in Guangzhou who use public STD clinics were found to have different sociodemographic characteristics, lower community engagement, and increased social cohesion, compared to MSM who use MSM-led clinics. Some comments and suggestions are as follows:

1.In the “Intrdution”, P.3”MSM-led clinics. Although same-sex behaviours were decriminalized in China in 1997, thereremains a lack of legal protection for MSM against discrimination, and same-sex marriage is illegal”, please provide references.

2.There was a big difference in sample size between the two groups, Public STD is only 63 participants and MSM-led is 175 participants. Was the sample size enough for this study?

3.In Table 1, it is suggested that the analysis of the two groups of sociodemographic characteristics, and psychological factors related to attend clinics, should not be placed in the same table.

4.Why did both types of clinic operate in the same facility in three cities (Guangzhou, Foshan, and Jiangmen)?

5.Was there any information collected on why the subjects chose to visit these institutions?

6.In Table 1, it is recommended to supplement subjects' condom use during sexual behaviors.

7.It is suggested that more detailed information on the two different types of institutions be added to the method.

6. PLOS authors have the option to publish the peer review history of their article (what does this mean?). If published, this will include your full peer review and any attached files.

Reviewer #1: No

Reviewer #2: No

---

## [Author Response · Author response to Decision Letter 0]

4 Sep 2024

Sociodemographic characteristics, community engagement and stigma among Men who have Sex with Men (MSM) who attend MSM-led versus public sexual health clinics: A cross-sectional survey in China (PONE-D-24-26827)

Dear Editor,

Thank you for the helpful editorial and reviewer comments and interest in accepting our manuscript for publication. 

We have revised the manuscript according to the requests suggested. Please find below our point-by-point response to the reviewer's comments.

Thank you.

Journal Requirements:

Response: We have ensured the revised manuscript meets PLOS ONE's style requirements.

2.Please include a complete copy of PLOS’ questionnaire on inclusivity in global research in your revised manuscript. Our policy for research in this area aims to improve transparency in the reporting of research performed outside of researchers’ own country or community. The policy applies to researchers who have travelled to a different country to conduct research, research with Indigenous populations or their lands, and research on cultural artefacts. The questionnaire can also be requested at the journal’s discretion for any other submissions, even if these conditions are not met. 

Please find more information on the policy and a link to download a blank copy of the questionnaire here: https://journals.plos.org/plosone/s/best-practices-in-research-reporting.

Please upload a completed version of your questionnaire as Supporting Information when you resubmit your manuscript.

Response: We have uploaded the questionnaire version used in the pilot study as supporting information. Please see the S1 File.

"National Institutes of Health (Grant number NIH NIAID R01AI158826)"

Response: This statement is correct. The funders had no role in study design, data collection and analysis, decision to publish, or manuscript preparation.

4. In the online submission form you indicate that your data is not available for proprietary reasons and have provided a contact point for accessing this data. Please note that your current contact point is a co-author on this manuscript. According to our Data Policy, the contact point must not be an author on the manuscript and must be an institutional contact, ideally not an individual. Please revise your data statement to a non-author institutional point of contact, such as a data access or ethics committee, and send this to us via return email. Please also include contact information for the third party organization, and please include the full citation of where the data can be found.

Response: We have updated the data statement to the University of North Carolina-Project China. The institutional email address ‘sesh@seshglobal.org’ has been 

5.Please review your reference list to ensure that it is complete and correct. If you have cited papers that have been retracted, please include the rationale for doing so in the manuscript text, or remove these references and replace them with relevant current references. Any changes to the reference list should be mentioned in the rebuttal letter that accompanies your revised manuscript. If you need to cite a retracted article, indicate the article’s retracted status in the References list and also include a citation and full reference for the retraction notice.

Response: We have double-checked all cited references to ensure they are complete and correct.

Reviewer 1:

This research article provides significant insights for the public health field by comparing the sociodemographic and psychosocial characteristics of Chinese men who have sex with men (MSM) seeking sexual health services at different types of clinics. The study targets the MSM population, a crucial group for the prevention and control of sexually transmitted diseases (STDs), which holds global significance in public health. Overall, this paper holds substantial research value in promoting the optimization of sexual health services for MSM in China. Some minor problems should be revised for the current version manuscript.

Response: Thank you

Introduction:

1.It is recommended to provide more background information on the existing literature and knowledge gaps regarding health-seeking behaviors and psychosocial factors among MSM in China.

2.Response: We have provided more background on the existing gap in the literature on lines 77-82 in the statement, "Some studies have investigated the psychological characteristics of MSM in China, and one found that MSM with and without high-risk homosexual behaviours in China have different social and psychological characteristics [14-16]. However, none of these studies have assessed the relationship between these psychological characteristics and healthcare seeking behaviors and testing sites among MSM in China to our knowledge. Additionally, most studies of healthcare seeking behaviour have focused on high income countries [17].

3.Methods: 

It is suggested to provide more details on the statistical analysis, including the variable selection process for the multivariate models and the handling of potential confounding factors.

Response: We have provided more details on the variable selection process for the multivariate models and the handling of potential confounders in the new Statistical analysis section on lines 128-140 in the statement:

“Statistical Analysis

A descriptive analysis was conducted to summarize the sociodemographic characteristics, sexual behaviors, and STD testing experience of participants. Sociodemographic data were analyzed for inter-group differences using Pearson's chi-squared test or Fisher's exact test. The association between psychosocial factors and clinic presentation was examined using univariate and multivariable logistic regression. Socio-demographics, psychosocial characteristics, and clinic type (public STD vs. MSM-led). Variables found to be marginally associated (a priori determined as p<0.20) with clinic type in crude bivariate analysis were included in a multivariable logistic regression model. No variable had >15% missing data, and variables with missing data were exempted from the regression models using a listwise deletion approach. Age, level of education, and marital status were controlled in the regression model as confounders of the psychosocial factors. The results were reported as adjusted odds ratio(aOR) with 95% confidence intervals (C.I) and p-values [17]."

Statistical analysis:

Further details should be provided on the statistical analysis, including the variable selection process for the multivariate models and the handling of potential confounding factors.

Response: We have provided more details on the variable selection process for the multivariate models and the handling of potential confounders in the new Statistical analysis section on lines 133-145 in the statement:

“Statistical Analysis

A descriptive analysis was conducted to summarize the sociodemographic characteristics, sexual behaviors, and STD testing experience of participants. Sociodemographic data were analyzed for inter-group differences using Pearson's chi-squared test or Fisher's exact test. The association between psychosocial factors and clinic presentation was examined using univariate and multivariable logistic regression. Socio-demographics, psychosocial characteristics, and clinic type (public STD vs. MSM-led). Variables found to be marginally associated (a priori determined as p<0.20) with clinic type in crude bivariate analysis were included in a multivariable logistic regression model. No variable had >15% missing data, and variables with missing data were exempted from the regression models using a listwise deletion approach. Age, level of education, and marital status were controlled in the regression model as confounders of the psychosocial factors. The results were reported as adjusted odds ratio(aOR) with 95% confidence intervals (C.I) and p-values."

4. Results:

5.Table 1, which compares the sociodemographic and psychosocial characteristics of MSM, is somewhat disorganized; partitioning it into two tables would be more appropriate.

Response: We agree with you and have partitioned the table into Tables 1 and 2. Thank you

6.Discussion:

7.It is recommended to discuss the potential limitations of the self-reported data and their impact on the validity of the study results.

Response: We have discussed the potential limitations of the self-reported data on lines 195-199 in the statement, "Furthermore, the data on sexual behaviours and psychosocial characteristics are self-reported and may be subject to recall and social desirability bias. Our investigation addressed sensitive questions that may have led to the misreporting of personal sexual risk and STD testing behavior data owing to social desirability bias. Therefore, our findings and conclusions should be interpreted cautiously."

Reviewer 2

This study was found that MSM in Guangzhou who use public STD clinics were found to have different sociodemographic characteristics, lower community engagement, and increased social cohesion, compared to MSM who use MSM-led clinics. 

Some comments and suggestions are as follows:

1. In the “Intrdution”, P.3”MSM-led clinics. Although same-sex behaviours were decriminalized in China in 1997, there remains a lack of legal protection for MSM against discrimination, and same-sex marriage is illegal”, please provide references.

Response: We have revised this phrase on lines 68-71 in the statement, " Although same-sex behaviours were decriminalized in China in 1997 and declassified as part of the Chinese Classification of Mental Disorders in 2001 [9, 10], there remains a lack of legal protection for MSM against discrimination, and same-sex marriage is illegal". We have also provided two references to support this statement.

Reference:

1.Tanner HM. The Offense of Hooliganism and The Moral Dimension of China's Pursuit of Modernity, 1979–1996. Twentieth-Century China. 2000;26(1):1-40. doi: 10.1179/tcc.2000.26.1.1.

2.Chen YF. Chinese classification of mental disorders (CCMD-3): towards integration in international classification. Psychopathology. 2002;35(2-3):171-5. Epub 2002/07/30. doi: 10.1159/000065140. PubMed PMID: 12145505.

2. There was a big difference in sample size between the two groups, Public STD is only 63 participants and MSM-led is 175 participants. Was the sample size enough for this study?

Response: We believe the sample size was enough for the study, as the generally recommended sample size for pilot trials ranges from 24 to 100 participants. 

Reference: 

1.Whitehead AL, Julious SA, Cooper CL, Campbell MJ. Estimating the sample size for a pilot randomised trial to minimise the overall trial sample size for the external pilot and main trial for a continuous outcome variable. Stat Methods Med Res 2016; 25(3): 1057-73.

2.Thorlund K, Thabane L, Mills EJ. Modelling heterogeneity variances in multiple treatment comparison meta-analysis--are informative priors the better solution? BMC Med Res Methodol. 2013; 13: 2.

3. In Table 1, it is suggested that the analysis of the two groups of sociodemographic characteristics, and psychological factors related to attending clinics, should not be placed in the same table.

Response: We agree with you and have partitioned the table into Tables 1 and 2. The revised Table 1 summarizes the sociodemographic characteristics of the participants, and Table 2 reports the findings of the logistic regression model, which shows the relationship between psychological factors and attended clinics. Thank you

4. Why did both types of clinic operate in the same facility in three cities (Guangzhou, Foshan, and Jiangmen)?

Response: In these sites, the MSM-led CBOs we partnered with had existing partnerships with the public STD clinics that enabled them to provide routine HIV testing, counselling, and other healthcare services to their local MSM communities at the facility. The clinics were also run by clinical staff identifying with the LGBTQ community as members or allies, creating an inclusive environment that respected and valued all patients, regardless of their potential distrust of other clinicians at the facility. Notably, most of the MSM-led clinics in other cities had partnerships with the local CDC and/or other local health facilities for laboratory testing and treatment referral purposes although they did not operate in the same facility buildings as these three.

5. Was there any information collected on why the subjects chose to visit these institutions?

Response: Unfortunately, this information was not collected.

6. In Table 1, it is recommended to supplement subjects' condom use during sexual behaviors.

Response: We have included the data on participants' condom use rates for the past 6 months in Table 1. 

7. It is suggested that more detailed information on the two different types of institutions be added to the method.

Response: We have stated on lines 103-104 that 'S1 Fig provides more details on the similarities and differences between the two clinic types'. Thank you

---

## [Editor Report · Decision Letter 1]

10 Sep 2024

Sociodemographic characteristics, community engagement and stigma among Men who have Sex with Men (MSM) who attend MSM-led versus public sexual health clinics: A cross-sectional survey in China

PONE-D-24-26827R1

Dear Dr. Tucker,

We’re pleased to inform you that your manuscript has been judged scientifically suitable for publication and will be formally accepted for publication once it meets all outstanding technical requirements.

Kind regards,

Geng-Feng Fu, Ph.D.

Academic Editor

PLOS ONE
---

## [Editor Report · Acceptance letter]

7 Oct 2024

PONE-D-24-26827R1 

PLOS ONE

Dear Dr. Tucker, 

I'm pleased to inform you that your manuscript has been deemed suitable for publication in PLOS ONE. Congratulations! Your manuscript is now being handed over to our production team.

Kind regards, 

on behalf of

Dr. Geng-Feng Fu 

Academic Editor

PLOS ONE